# Fabrication and Application of Black Phosphorene/Graphene Composite Material as a Flame Retardant

**DOI:** 10.3390/polym11020193

**Published:** 2019-01-22

**Authors:** Xinlin Ren, Yi Mei, Peichao Lian, Delong Xie, Weibin Deng, Yaling Wen, Yong Luo

**Affiliations:** 1Faculty of Environmental Science and Engineering, Kunming University of Science and Technology, Kunming 650500, China; ren8877@126.com; 2Faculty of Chemical Engineering, Kunming University of Science and Technology, Kunming 650500, China; cedlxie@kmust.edu.cn (D.X.); weibindengg@163.com (W.D.); wenyaling66@126.com (Y.W.); luoyong0327@126.com (Y.L.); 3The Higher Educational Key Laboratory for Phosphorus Chemical Engineering of Yunnan Province, Kunming University of Science and Technology, Kunming 650500, China

**Keywords:** black phosphorene, graphene, flame retardant, mechanical performance, waterborne polyurethane

## Abstract

A simple and novel route is developed for fabricating BP-based composite materials to improve the thermo-stability, flame retardant performances, and mechanical performances of polymers. Black phosphorene (BP) has outstanding flame retardant properties, however, it causes the mechanical degradation of waterborne polyurethane (WPU). In order to solve this problem, the graphene is introduced to fabricate the black phosphorene/graphene (BP/G) composite material by high-pressure nano-homogenizer machine (HNHM). The structure, thermo-stability, flame retardant properties, and mechanical performance of composites are analyzed by a series of tests. The structure characterization results show that the BP/G composite material can distribute uniformly into the WPU. The addition of BP/G significantly improves the residues of WPU in both of TG analysis (5.64%) and cone calorimeter (CC) test (12.50%), which indicate that the BP/G can effectively restrict the degradation of WPU under high temperature. The CC test indicates that BP/G/WPU has a lower peak release rate (PHRR) and total heat release (THR), which decrease by 48.18% and 38.63%, respectively, than that of the pure WPU, respectively. The mechanical analysis presents that the Young’s modulus of the BP/G/WPU has an increase of seven times more than that of the BP/WPU, which indicates that the introduce of graphene can effectively improve the mechanical properties of BP/WPU.

## 1. Introduction

Black phosphorene (BP) has triggered intense studies in recent years due to its two-dimensional (2D) structure and distinct physical and chemical properties [1,2], which makes BP a promising candidate for electronic, photonic, medical and energy storage devices [3,4,5,6,7,8,9]. We have also reported that the BP can be applied as a flame retardant for waterborne polyurethane (WPU) due to its outstanding flame-retardant properties [10]. However, as the same problem as some other inorganic retardants [11], the addition of BP makes the mechanical performance of the polymer decrease, for example, the WPU films become crispy and fragile, which would constrict its application in the flame retardant field.

Traditionally, numerous efforts have been made to reinforce the mechanical performances of the polymeric materials through adding some nanofillers [12,13,14,15]. Especially, it is reported that the graphene can distribute uniformly into the matrix materials and qualify a good compatibility with polymers, which can effectively improve the mechanical performance of polymers [16,17,18,19]. In addition, graphene can be also applied as a flame retarding material because it exhibits the thermo-stability, as well as good environmental acceptance [20,21,22,23]. Thus, graphene can simultaneously improve the fire restriction and mechanical performance of the polymers [24,25]. However, the flame retardant effect of the graphene only works in the condensed phase, resulting that it cannot play a perfect flame retardant role [26,27]. Taking into consideration the flame retardant and mechanical performance of graphene and BP for polymers, the combination of them seems to be very meaningful.

Additionally, a single type of element cannot always have the best degradation restriction and flame retardant properties, leading that the composite materials are the research trend in flame retardant field [28,29]. Therefore, the graphene may be combined with the BP to improve the flame retardant performance, as well as mechanical performance of polymers. In this study, we will fabricate the composite material of BP and graphene through a simple method by high-pressure nano homogenizer machine (HNHM). The prepared composite materials will be used to fill into WPU to investigate the thermal decomposition, flame retardant, and mechanical properties.

## 2. Materials and Methods

### 2.1. Materials

The black phosphorus was prepared by the mineralization method [30]. After a specific heating and cooling process, the red phosphorus (RP) was transformed in to black phosphorus with the help of iodine and tin in a quartz tube under argon atmosphere. In order to remove the residual mineralizer, the black phosphorus was first washed by toluene, and then washed with water and acetone. The involved RP, iodine, tin, toluene and acetone were analytical pure. The graphite with the particle size lower than 30 μm was purchased from Shanghai Huayi Group Huayuan Fine Chemicals Co., Ltd. (Shanghai, China). The WPU latex with a solid content of 25 wt% was purchased from Anhui Huatai New Material Co., Ltd. (Hefei, China). The involved water was deionized water.

### 2.2. Fabrication of Black Phosphorene/Graphene (BP/G) Composite Materials

The black phosphorus was ground for 2 h into powders, and then 1 g of the powders were added into a conical flask with 500 mL of the deionized water. One gram of the graphite particles was weighed precisely and filled into the flask. A small amount (0.2 g) of sodium dodecyl benzene sulfonate (SDBS) was added to promote the distribution of black phosphorus and graphite into water. Then the dispersion liquid of black phosphorus and graphite particles were pretreated under a working ultrasonic bath (50 Hz, 200 W) for 2 h with controlling the temperature lower than 30 °C. Afterwards, the mixed dispersion liquid was poured into the high-pressure nano homogenizer machine (HNHM, AH-BASIC I, ATS Engineering Inc., Suzhou, China). Then the homogenizing valve was activated and slowly increased to a pressure of 1000 MPa. The strong sheer force of the machine could provide enough power to exfoliate the black phosphorus and graphite, and the schematic diagram to synthesize BP/G composite material is shown as Figure 1. Then the outputted dispersion liquid was re-poured into the machine. After repeating this process 30 times, the dispersion liquid was collected and centrifuged at 3500 rpm for 10 min by a centrifugal machine (TGL-16C, Shanghai Anting Scientific Instrument Factory, Shanghai, China). Finally, the supernatant liquid containing BP/G was collected and condensed by suction filtration with most of SDBS were removed. An argon atmosphere should be used to prevent the oxidation of phosphorene in the whole experimental process.

### 2.3. Preparation of Black Phosphorene/Graphene/WPU (BP/GWPU) Composite Materials

WPU latex and BP/G dispersion were added into a beaker. After stirring for a few minutes, the beaker was sealed with argon atmosphere filling. Then the mixed suspension liquid was blended by the ultrasonic device for 2 h with the temperature lower than 30 °C by using an ice bath. Then the obtained suspension liquid was poured into a plate with the size of 120 × 120 mm and dried into a vacuum drying oven at 22 °C. After it was dried completely, the BP/G/WPU membrane was formed.

### 2.4. Analytical Test

#### 2.4.1. Structure Characterizations

Transmission electron microscopy (TEM, Philips CM100, Amsterdam, the Netherlands) was conducted to observe the BP/G composite material at an acceleration voltage of 100 kV.

X-ray diffraction device (XRD, PANalytical Empyrean, Almelo, Netherlands) were conducted to analyze the BP, graphene and BP/G powders, as well as the G/WPU and BP/G/WPU membranes, respectively.

The scanning electron microscopy (SEM, Bruker Nano, Bruker, Karlsruhe, Germany) was taken on the fracture surfaces of polymeric membranes by liquid nitrogen. The fracture surfaces were coated with a layer of gold prior to testing. In order to present the distribution of the additives in the polymers, elemental mapping tests were conducted on another two specimens without coating of gold layer. In addition, The EDS on the same device of SEM was applied to determine the additive amount of BP.

#### 2.4.2. Thermal Properties Measurement

Thermogravimetric analysis (TGA) was performed on a thermal analyzer (NETZSCH STA449F3, NETZSCH, Selb, Germany). The heating rate was 10 °C/min from 40 to 800 °C and the gas flow rate was 80 mL/min in the nitrogen atmosphere.

#### 2.4.3. Flammability Property Measurement

Cone calorimetry (CC) was performed on a PHINIX combustion calorimeter (PX-07-007, Suzhou Phoenix Quality Inspection Instrument Co., Ltd, Suzhou, China) at a heat flux of 35 kW/m^2^. The specimens are made into a square with the size of 100 × 100 × 3 mm, and then wrapped in a piece of aluminum film.

#### 2.4.4. Mechanical Test

The tensile tests were carried out on an electronic universal testing machine (CMT6104, MTS Systems Corporation, Shanghai, China) with a deformation rate of 55 mm/min according to ASTM D638-14.

## 3. Results and Discussion

After the BP/G materials being synthesized and dried in a vacuum freeze drier, X-ray diffraction (XRD) was carried out on the BP, G, and BP/G powders, respectively, which are shown in the Figure 2. From Figure 2. we can see that the diffraction peaks of BP/G composite material correspond to both of diffraction peaks of the BP and G, indicating that the BP/G composite material is comprised of BP and G. However, the characteristic reflections of BP become weaker in BP/G, this may be due to the breakup of some P−P bonds for generation of the P−C bonds. As a result, BP will become more stable due to the generation of stable P−C bonds in BP/G composite materials. This phenomenon is consistent with the literatures [31,32,33], which were reported that there had been formed P-C bonds between the two type of materials, and they could eliminate the disadvantage of instability of BP in ambient environment.

Raman spectra was also conducted on the BP, G, BP/G composite material. As shown in Figure 3a, the BP flakes show three characteristic peaks, which are the Raman peaks of A_g_^1^, B_g_^2^, and A_g_^2^, respectively. The G flakes show two characteristic peaks. In the enlarged image of the Raman shift ranging from 300 to 500 cm^−1^ (see Figure 3b), the A_g_^1^, B_g_^2^, and A_g_^2^ of BP are presented at the 360 cm^−1^, 437 cm^−1^ and 465 cm^−1^, respectively. However, the corresponding three peaks of BP/G composite material prepared by HNHM have little offsets, which implies that the structure of BP has been changed a little. That is due to the generation of P-C bonds after being dealt with by the HNHM, leading to some different vibrations in-plane and out-plane [34].

For further testify the structure, the TEM and SEM were used to observe the microstructure, and the results are shown in Figure 4. As we can see that the BP and graphene nanosheets have a thin layered structure with the size of several hundred nanometers. A high-resolution transmission electron microscopy (HRTEM) image of the sample is shown in Figure 4c, and we can see that the BP has good crystalline structure. The SEM with an elemental mapping were tested on a dried powder sample, which can be seen in Figure 4d–f. The elemental phosphorus and carbon present uniformity in this image, which indicates that the phosphorene and graphene are exfoliated by the HNHM. The results of TEM, XRD, and Raman tests confirm that the BP and G sheets has generated few P-C bonds, which is benefit for the application of this type of composite material.

Graphene and BP/G were added into the WPU matrix to form G/WPU and BP/G/WPU membranes. The XRD were conducted on the membranes, which are shown in Figure 5a. Compared with the Figure 2, we can see that the diffraction peaks of BP/G/WPU is comprised of the peaks of BP, G and BP/G, which indicates that BP/G composite materials has been added into the WPU matrix successfully. Similarly, the diffraction peaks of G/WPU show that G has been added into the matrix as well. Figure 5b shows the Raman spectra of the pure WPU, G/WPU and BP/G/WPU, it also testify that the BP/G/WPU show the Raman peaks of BP and G at the same time, indicating that the BP/G composite material has been added into the WPU polymers.

In order to analyze the additive amount of BP/G in the BP/G/WPU membrane, we firstly determine the ratio of P/C in BP/G composite material by using TG device under nitrogen gas. From the TG curves (Figure 6) we can see that the graphene is very stable, which changes little with the increase of temperature. Whereas, the TG curve of BP/G composite material has a significantly weight loss after 400 °C, this is because that the phosphorus is volatilized under a high temperature and the residue is carbon [35]. At the temperature of 800 °C, the BP/G composite material has a residue of 42%, which indicates that the graphene is approximately 42% in the BP/G composite material. Therefore, the ratio of P/C in the BP/C composite material is 1.38:1, although the initial ratio of P/C is 1:1. Combining with the EDS result (Figure 7f) of BP/G/WPU membrane that displays that the content of P in BP/G/WPU membrane is 2.06%, the additive amount of BP/G composite material in BP/G/WPU membrane can be calculated as 3.55%. The results are also shown in Table 1.

Phosphorus mapping and the corresponding EDS of a region in the membranes were conducted on the surface (see Figure 7) to further observe the distribution of the additive materials. From the figure, we can see that there are little red spots in the pure WPU (see Figure 7a) and G/WPU (see Figure 7c), whereas, the corresponding EDS results indicate that there is scarcely any phosphorus. The reason is may be the little phosphorus element existed around the environment or the interference signal from the peak near the position of phosphorus. The phosphorus element in Figure 7e,f (see as the red sheet areas) presents a homogeneous, independent, and layered structure, which shows that the BP/G composite materials have a uniform distribution into the specific zone.

The microstructure of the pure WPU (Figure 8a,b), G/WPU (Figure 8c,d), and BP/G/WPU (Figure 8e,f) were observed using SEM on the brittle fractured surfaces by liquid nitrogen. The images of pure WPU shows homogeneous and without any additive particles due to the typical fracture behavior of homogeneous material. Compared with the pure WPU, the G/WPU composite material shows more white sheets, which are the additive graphene sheets. As we can see that the graphene sheets distribute homogeneously in WPU. The images of BP/G/WPU show that the BP/G materials are much bigger and rougher than graphene sheets, indicating that the black phosphorene sheets and graphene sheets are combined together.

Study on the thermal stabilities in an anoxic atmosphere may predict the decomposition behaviors in a real fire scenario. The TGA and resulted DTG curves of the BP/G composite material in a matrix are shown in the Figure 9. The curves show two major mass loss stages due to the hard segment and soft segment in polyurethane, and the hard segment is more prone to thermal decomposition than the soft segment [36]. The first stage takes place at the temperature lower than 380 °C, which is mainly caused by the breakage of urethane bonds in the amine and isocyanate. The second stage ranges from 380 to 500 °C, which is assigned to the decomposition of residual polyols. The initial decomposition temperature (T5%) of BP/G/WPU is 222.3 °C, which is a little lower than that of pure WPU (236.7 °C) and higher than that of G/WPU (212.5 °C). The reason is that the BP forms into phosphorous acid under high temperature, and then the phosphorous acid promotes the thermal decomposition of polyurethane [37]. What’s more, the additive graphene may capture oxygen in the WPU, leading to the decrease of the initial decomposition temperature of G/WPU. Once the membranes begin to decompose, the weight loss rate of BP/G/WPU becomes the slowest, which indicates that the BP/G can promote the stability of polymers. Furthermore, the residue char of WPU, G/WPU and BP/G/WPU are 0.96%, 3.05%, 5.64%, respectively, which also indicates that the BP/G composite material can effectively improve the stability of polymers. The reason is that the BP plays a leading role due to the formation of phosphorous acid, which can promote the removal of oxygen in polymers and the generation of a char layer.

The three type of membranes were tested by CC, which could be applied to simulate a real fire scenario. The heat release rate (HRR) curves and total heat release (THR) curves are shown as Figure 10a,b. And the detailed data are shown in Table 2. The peak value of the HRR (PHRR) is a key parameter to manifest the propagation ability of the fire and ignition ability of the materials. The PHRR of the WPU, G/WPU and BP/G/WPU are 454.3, 358.0, and 235.4 kW/m^2^, respectively. These data show that the pure WPU is easy to set on fire and lead to the “flashover” phenomenon, however, the addition of graphene can make the PHRR decrease by 21.2% and the addition of BP/G can decrease PHRR by 48.18% than that of the pure WPU. This is because their special layered structure of the graphene and BP have an energy barrier effect during the pyrolysis process, which can prevent the materials further contacting with the oxygen and heat transferring to the other places. In addition, BP can play a key role because they can form into PO radicals and diffuse around in the surrounding gas, which can react with the H or OH radicals generated by polymers under the burning condition, consequently reduce the energy of the flame [38]. Phosphorus may also form into phosphoric acid under high temperature, which can effectively promote the polymers to produce carbon layer, consequently prevent the transport of oxygen to the burning zone and weaken the flame [39,40]. The THR curves (Figure 10b) indicate that the pure WPU releases a large amount of heat. Compared with the pure WPU and G/WPU, the BP/G/WPU release the least of heat with the value of only 51.68 MJ/m^2^, which decreases by 38.63% and 34.00%, respectively. The time to ignition (TTI) and time to PHRR (TPHRR) values of the membranes are shown in the Table 2. From Table 2, we can see that BP/G/WPU has the biggest TTI comparing to the pure WPU and G/WPU, which indicates that the BP/G makes the polymer matrix difficult to ignite. What’s more, TPHRR of BP/G/WPU is the earlier to arrive. This result shows that the duration of combustion of BP/G/WPU becomes shorter due to the addition of BP/G. That is to say once the material is set on fire, the fire will weaken soon. The CO release (Figure 10c) and the CO_2_ release (Figure 10d) curves indicate that BP/G/WPU release more CO and less CO_2_. Even so, the total carbon emission of BP/G/WPU is the lowest, which indicates that the BP/G can make the polymer incomplete combustion and most of the carbon are retained in the residues.

The residues after CC tests of the membranes can also speak volume for the ability of BP/G composite materials to restrain heat risk. From the Table 3, we can see that the BP/G/WPU has a residue of 12.5% after the combustion under high temperature, while the WPU has scarcely any residues and the G/WPU has a residue of 4.0%, which are much lower than the BP/G/WPU. This phenomenon can also be confirmed by the images of residues (see Figure 11). As we can see, the pure WPU has nothing left after combustion, the G/WPU shows some scattered and thin char layer, whereas the BP/G/WPU produces more residues, which are denser and thicker, some even stacking together into a bulk. These results imply that the BP/G composite material has a better ability of flame retardant than the single BP.

Mechanical properties of the composites were investigated using tensile tests. Their results are shown in Figure 12 and Table 3. From the load/displacement curves (Figure 12), we can see that the load at break and the maximal load of BP/WPU are decreased in comparison of the pure WPU, which indicates that the mechanical performance becomes poor due to the addition of BP. The load at break and the maximal load decrease in the order of BP/G/WPU > WPU > BP/WPU, indicating that the introduction of the graphene has played a key role to enhance the mechanical performance of the WPU and BP/WPU. From the Table 3, we can see that the calculated Young’s modulus of the BP/WPU shows a sharp decrease than that of the other samples. This phenomenon is consistent with the load/displacement curves and the previous literatures [41,42,43]. The Young’s modulus of the BP/G/WPU has an increase of seven times more than that of the BP/WPU due to the introduction of graphene. Additionally, the data of the ultimate strength of the samples show the same trend of Young’s modulus. The ultimate strength of the BP/WPU shows a decrease of 26.39% relative to the pure WPU. As seen from the table, the data of the elongation at break are on the contrary to the Young’s modulus and ultimate strength. The elongation at break of BP/WPU has a significantly increase, reaching to 1231.51%, which indicates that the BP increases the malleability of polymers, whereas decreases the tensile strength. Overall, the BP has a negative effect and the graphene has a positive effect to the mechanical performance of the polymer. These results indicate that the introduction of graphene can effectively improve the mechanical performance of BP/WPU composite material. In addition, the BP/G composite material can effectively improve the flame retardant properties without damaging the mechanical performances of the WPU, which is a promising functional accessory ingredient material for polymers.

## 4. Conclusions

We have designed and fabricated a type of BP/G composite material by a high-pressure nano homogenizer, which could form P-C bonds in the edge of the two type of nanosheets. A total of 3.55% of the BP/G was added into the WPU to investigate the flame retardant and mechanical performances. The analysis results show that the BP/G composite material distributes uniformly into the WPU with a layered structure. The BP/G significantly restrict the degradation and heat release of the WPU. Compared with the WPU, BP/G/WPU has a lower PHRR (235.4 kW/m^2^) and THR (51.68 MJ/m^2^) in the CC test, which decreases by 48.18% and 38.63%, respectively. In addition, BP/G/WPU has a higher residue char than WPU and G/WPU in both of TG analysis (5.64%) and CC test (12.50%). The mechanical analysis presents that the Young’s modulus of the BP/G/WPU has an increase of seven times more than that of the BP/WPU, which indicates that the introduction of graphene can effectively improve the mechanical properties of the single BP/WPU. This work develops a simple and novel route for fabricating BP-based composite materials to improve the thermos-stability, flame retardant performances and mechanical performances of polymers, which would broaden the applications of BP in flame retardant field.

## Figures and Tables

**Figure 1 polymers-11-00193-f001:**
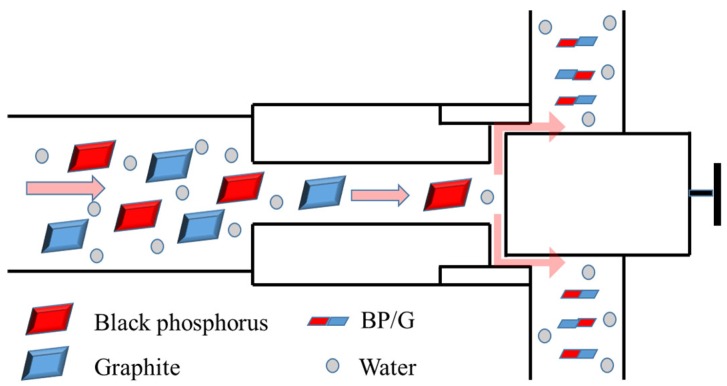
Schematic diagram to synthesize BP/G composite material by a high-pressure nano homogenizer machine.

**Figure 2 polymers-11-00193-f002:**
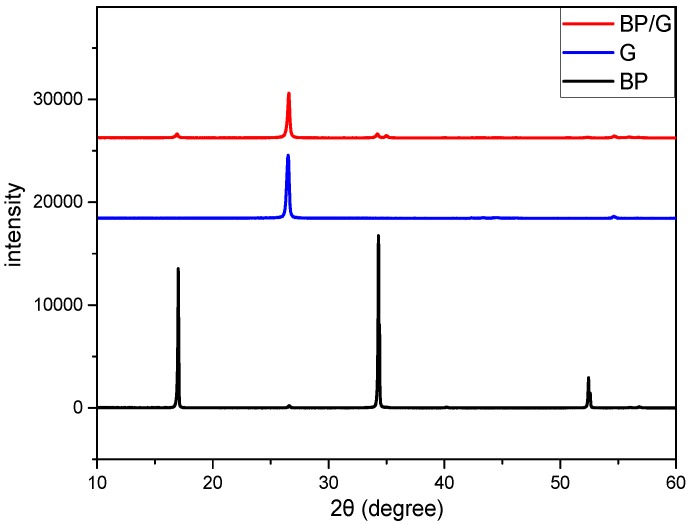
XRD spectra of BP, G, BP/G composite material.

**Figure 3 polymers-11-00193-f003:**
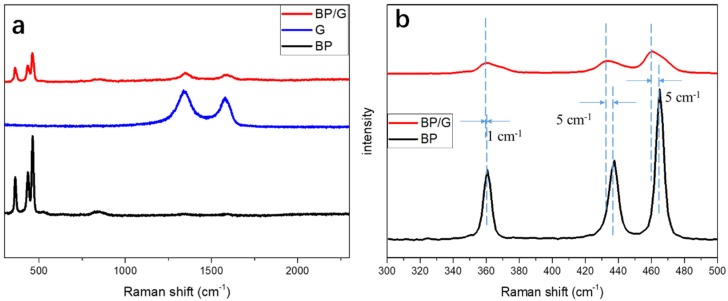
(**a**) Raman spectra of BP, G, BP/G composite material; and (**b**) the enlarged Raman spectra of BP and BP/G.

**Figure 4 polymers-11-00193-f004:**
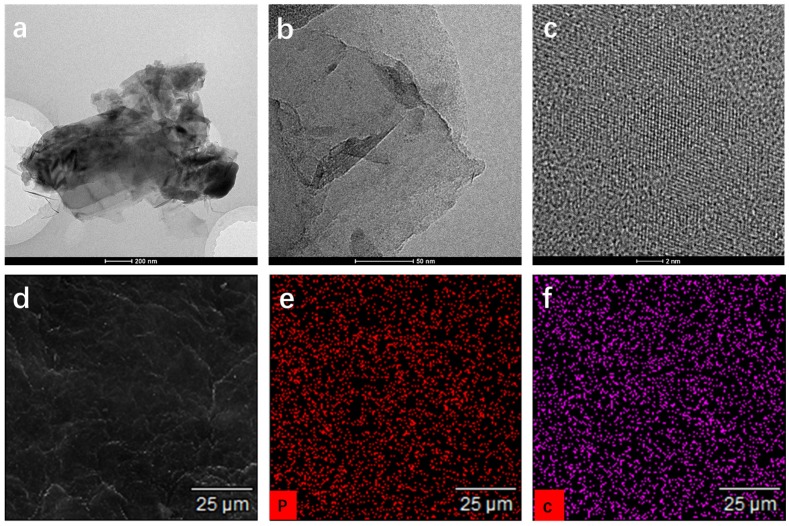
(**a**) TEM image of the BP/G composite materials; (**b**) enlarged TEM image of the BP/G composite materials; (**c**) HRTEM image of the BP/G composite materials; (**d**) SEM image of the BP/G composite materials; (**e**) phosphorus mapping of the BP/G composite materials; and (**f**) carbon mapping the BP/G composite materials.

**Figure 5 polymers-11-00193-f005:**
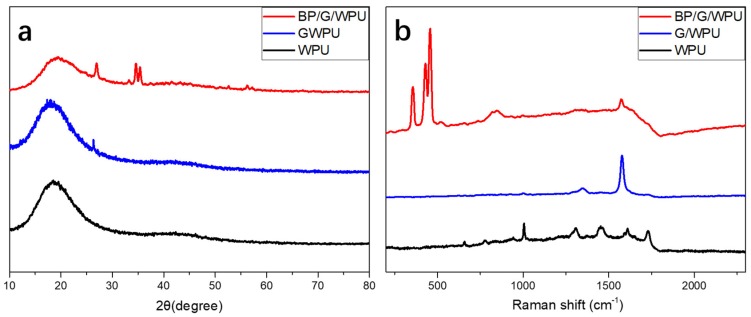
(**a**) XRD spectra of WPU, G/WPU and BP/G/WPU; and (**b**) Raman spectra of WPU, G/WPU, and BP/G/WPU.

**Figure 6 polymers-11-00193-f006:**
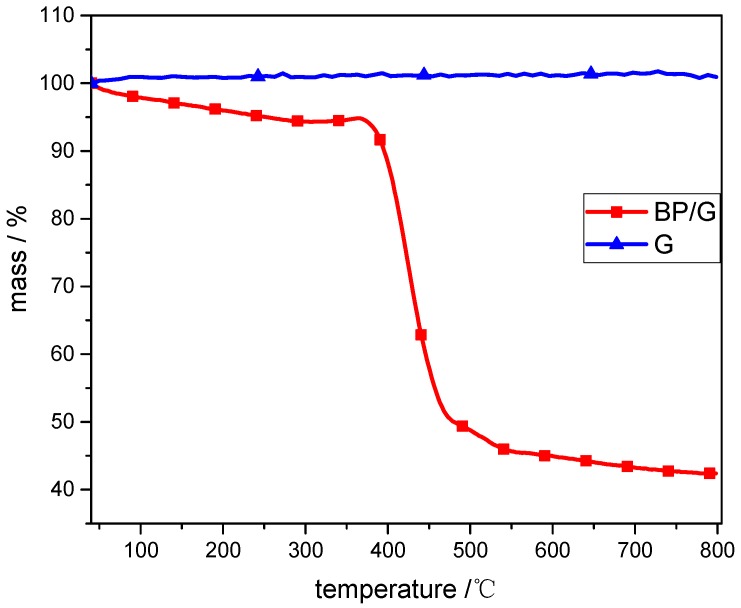
TG image of BP/G composite material and graphene, indicating the BP accounts of 58% of the BP/G composite material.

**Figure 7 polymers-11-00193-f007:**
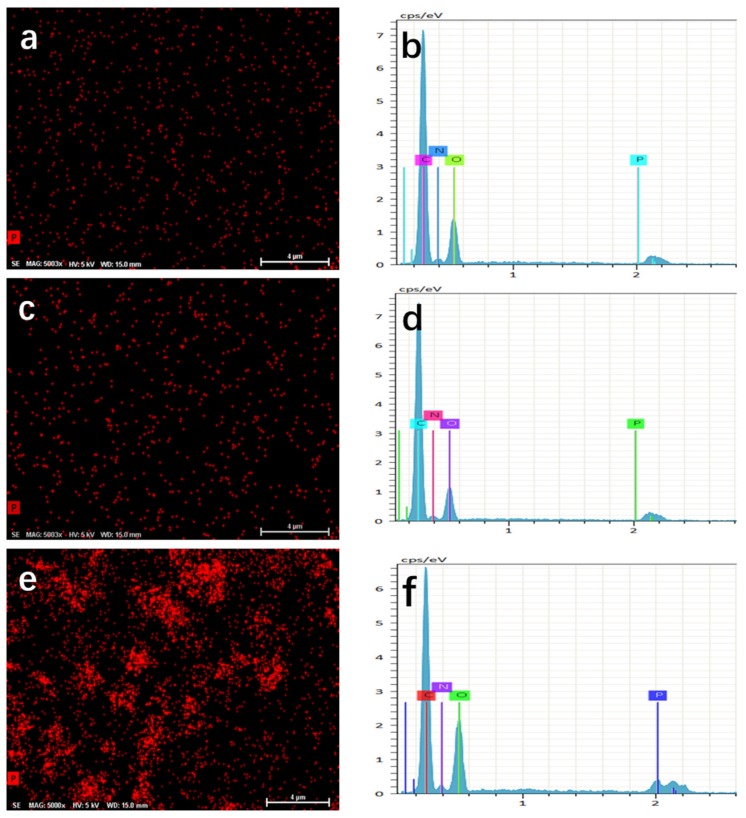
(**a**) phosphorus mapping of a region in pure WPU; (**b**) EDS test of the pure WPU; (**c**) phosphorus mapping of a region in pure BP/GWPU; (**d**) EDS test of the G/WPU; (**e**) phosphorus mapping of a region in G/WPU; (**f**) phosphorus mapping of a region in BP/G/WPU.

**Figure 8 polymers-11-00193-f008:**
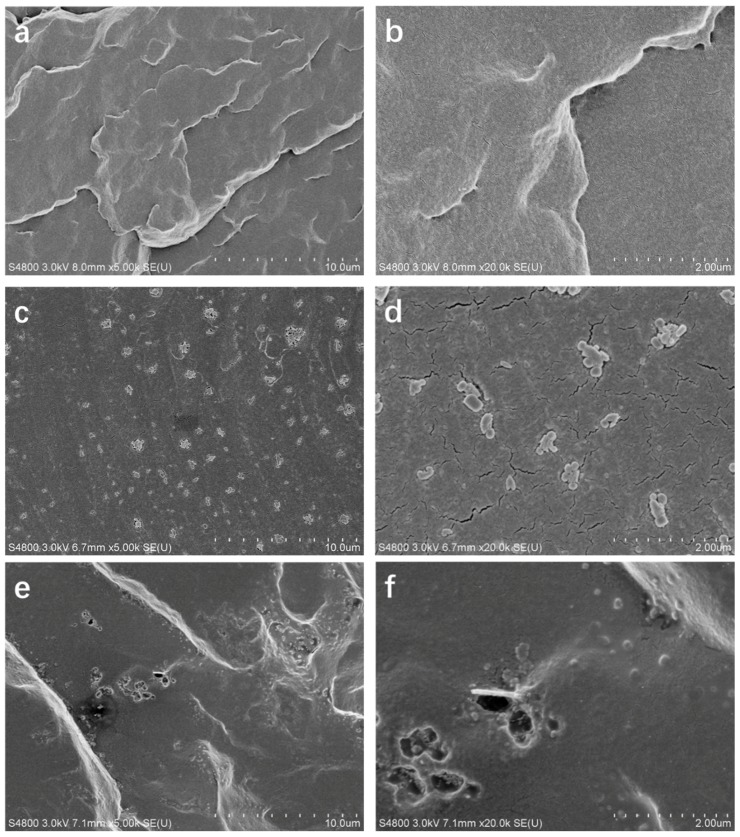
(**a**) SEM image of WPU; (**b**) enlarged SEM image of WPU; (**c**) SEM image of G/WPU; (**d**) enlarged SEM image of G/WPU; (**e**) SEM image of BP/G/WPU; and (**f**) enlarged SEM image of BP/GWPU.

**Figure 9 polymers-11-00193-f009:**
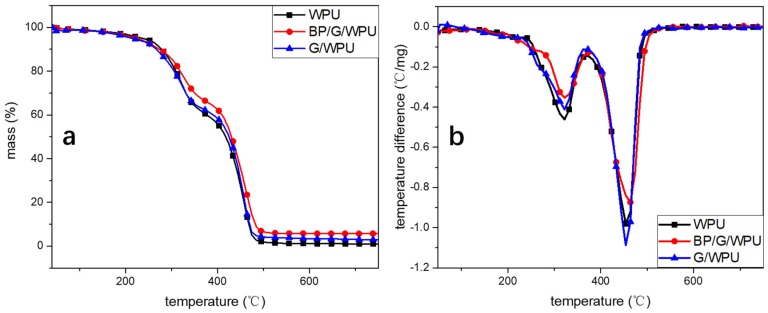
(**a**) the TGA curves; and (**b**) the resulted DTG curves.

**Figure 10 polymers-11-00193-f010:**
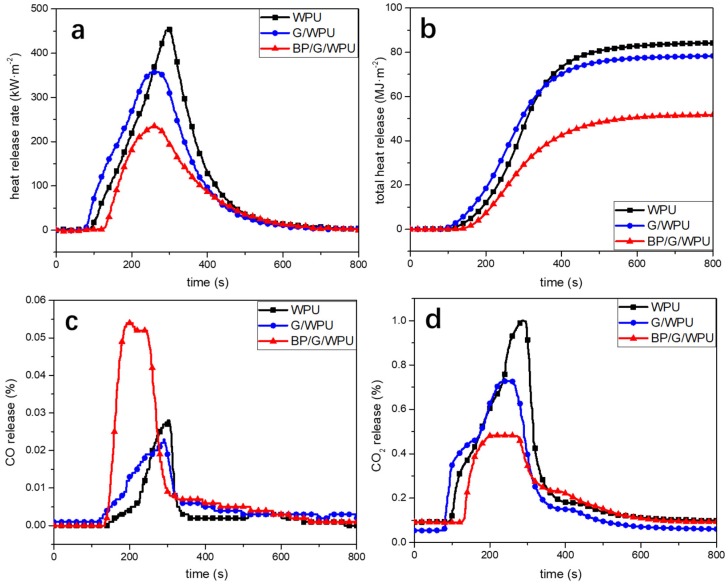
Cone calorimeter test of PLA composites at the concentration of 10 wt%: (**a**) HRR curves; (**b**) THR curves; (**c**) CO release curves; (**d**) CO_2_ release curves.

**Figure 11 polymers-11-00193-f011:**
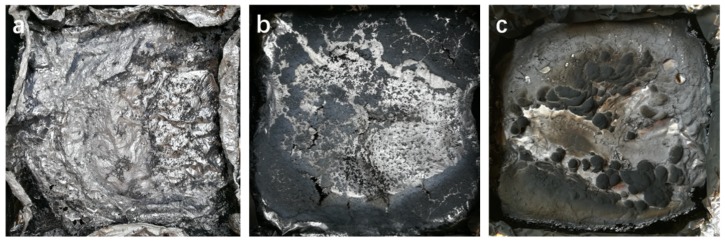
Images of the residues after the CC test: (**a**) residue of WPU; (**b**) residue of G/WPU; and (**c**) residue of BP/GWPU.

**Figure 12 polymers-11-00193-f012:**
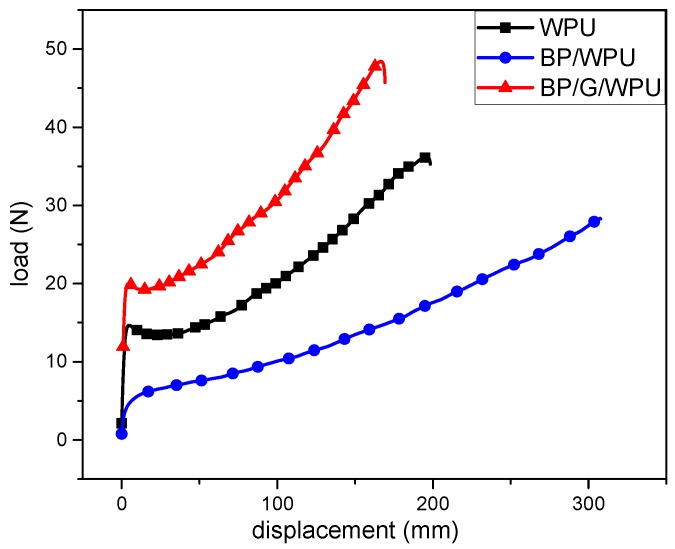
Load/displacement curves for the pure WPU, BP/WPU, G/WPU and BP/G/WPU.

**Table 1 polymers-11-00193-t001:** Additive amount of BP/G in BP/G/WPU membrane.

Items	Initial Ratio of P/C	Ratio of P/C in BP/G Composite Material	Content of P in BP/G/WPU Membrane (%)	Additive Amount of BP/G in BP/G/WPU Membrane (%)
Value	1	1.38	2.06	3.55

**Table 2 polymers-11-00193-t002:** The CC data of WPU, G/WPU, and BP/G/WPU.

Sample	TTI (s)	PHRR (kW/m^2^)	T-PHRR (s)	THR (MJ/m^2^)	Residue (%)
WPU	65	454.3	297	84.21	0
G/WPU	54	358.0	257	78.30	4.0
BP/G/WPU	101	235.4	259	51.68	12.5

**Table 3 polymers-11-00193-t003:** The mechanical properties of Young’s modulus, elongation at break and ultimate strength recorded from the tensile tests analysis.

Sample	Young’s Modulus (MPa)	Elongation at Break (%)	Ultimate Strength (MPa)
WPU	76.24	793.45	14.7
BP/WPU	6.18	1231.51	10.82
BP/G/WPU	48.44	677.09	11.72

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
