# Peer review of "Fabrication and Application of Black Phosphorene/Graphene Composite Material as a Flame Retardant"

_polymers, 2019, doi:10.3390/polym11020193_

Reviewer 1 Report

This manuscript described the effect of black phosphorene/graphene composite as a flame retardant in waterborne polyurethane membrane. This manuscript is in general well-organized and merits publication. However, there are several issues need to be addressed:

1. The English technical writing can be improved. There are several grammar mistakes, especially misuse of the verbs and missing propositions, that affected the quality of this manuscript.

2. It’s not very convincing of the existence of the P-C bond in the TEM just because BP and graphene are contacting through the edges. Also, from the TGA, at what temperature does the P-C bond break?

3. The author should try to provide a clear P-mapping.

4. It is likely that the residual mass in TGA was just the carbon char content of graphene, it is not convincing that “BP/G composites material can effectively improve the stability of polymers”.

Author Response

Reviewer 1:

This manuscript described the effect of black phosphorene/graphene composite as a flame retardant in waterborne polyurethane membrane. This manuscript is in general well-organized and merits publication. However, there are several issues need to be addressed:

1. The English technical writing can be improved. There are several grammar mistakes, especially misuse of the verbs and missing propositions, that affected the quality of this manuscript.

Response: We are very grateful for your comments. We have thoroughly reorganized and revised the manuscript and carefully minimized the language errors.

2. It’s not very convincing of the existence of the P-C bond in the TEM just because BP and graphene are contacting through the edges. Also, from the TGA, at what temperature does the P-C bond break?

Response: Thank you for your instructive suggestions. We have recognized that the TEM cannot directly present the existence of the P-C bond after considering your viewpoint and further communicating with the professional analysis tests persons. Thus, we have added the SEM image and corresponding elemental mapping in Figure 4 for analyzing the structure. From the figure, we can see that the elemental P and C distribute uniformly, which indicates that the P and C may intertwine together and form P-C bonds. In addition, the characterization results of XRD and Raman (Figure 2 and Figure 3) show that there exist few P-C bonds, which can be confirm from the literature [31-34], as a result, the existence of P-C bond can be testified. However, there is no significant signal shown in the TGA results, the reason may be that the content is too low to be determined in a TGA.

3. The author should try to provide a clear P-mapping.

Response: We have added the corresponding EDS images in the Figure 8. We can see that there are little red spots in the pure WPU and G/WPU, whereas, the corresponding EDS results indicate that there is no phosphorus. The reason is may be the little phosphorus element existed around the environment or the interference signal from the peak near the position of phosphorus.

4. It is likely that the residual mass in TGA was just the carbon char content of graphene, it is not convincing that “BP/G composites material can effectively improve the stability of polymers”.

Response: Thank you for the comment, this is a good question. In the manuscript, Table 1 shows that the additive amount of graphene is 1.38%, while the residual mass in TGA of BP/G/WPU is 5.64%, which has a major increase, which indicates that it can effectively improve the stability of polymers.

Reviewer 2 Report

In this manuscript, the authors reported the fabrication of BP-based composite materials for improving the thermos-stability, flame retardant performances and mechanical performances of polymers. These properties of WPU nanocomposites has been investigated in detail. Finally, the flame retardant enhancement mechanism was proposed. However, some issues should be addressed before publication.

1.     As is well known, black phosphorene is unstable in aqueous system. What about the BP-based WPU nanocomposites after drying the BP-based aqueous suspension?

2.     Page 4. The authors should add references to support your description in the Raman results discussion.

3.     Page 4. The authors should add TEM elemental mapping to confirm which nanosheets are BP.

4.     Page 5. TG image of BP/G composite material indicates the BP accounts of 58% of the BP/G composite material. How do you calculate the content of BP? The authors should provide the TG results of respective graphene and BP from room temperature to 800 oC. BP/G shows gradual weight loss in the temperature range of RT-750 oC.  

5.     Page 6. The authors declared that the fractured surface of WPU is smooth,but I think this is not the case. Please revise the description.

6.     Page 7. The authors contributed little red spots in the pure WPU and G/WPU to the test error of the test or phosphorus element existed around the environment. But I think that phosphorus mapping of a region in G/WPU is obvious compared to pure WPU. How do you explain the results.

7.     Page 9. The char residues for G/WPU is 0? Graphene has high char residues at high temperature. At least graphene is left after combustion.

8.     The paper is hard to read due to the poor language, including plenty of grammar errors. Please revise the language thorough the text. 

Author Response

Reviewer 2:

In this manuscript, the authors reported the fabrication of BP-based composite materials for improving the thermos-stability, flame retardant performances and mechanical performances of polymers. These properties of WPU nanocomposites has been investigated in detail. Finally, the flame retardant enhancement mechanism was proposed. However, some issues should be addressed before publication.

1. As is well known, black phosphorene is unstable in aqueous system. What about the BP-based WPU nanocomposites after drying the BP-based aqueous suspension?

Response: The obtained nanocomposites suspension liquid was dried into a vacuum drying oven at 22 ℃. Most of the air are pumped out and the temperature is low, thus, the BP cannot be oxidized. Thus, it can maintain the original appearance. This result can be confirmed by the figure 5 and figure 8.

2. Page 4. The authors should add references to support your description in the Raman results discussion.

Response: Thank you for your kind reminding. The reference [34] has been added in the revised manuscript.

3.Page 4. The authors should add TEM elemental mapping to confirm which nanosheets are BP.

Response: Thank you for your suggestions. We have added the High resolution transmission electron microscopy (HRTEM) image of the BP/G sample, which can indicate the good crystallinity of the BP. In addition, we have added the SEM image and corresponding elemental mapping in Figure 4 for analyzing the structure. From the figure, we can see that the elemental P and C distribute uniformly, which indicates that the P and C may intertwine together. We have also improved the description in the revised manuscript.

4. Page 5. TG image of BP/G composite material indicates the BP accounts of 58% of the BP/G composite material. How do you calculate the content of BP? The authors should provide the TG results of respective graphene and BP from room temperature to 800 . BP/G shows gradual weight loss in the temperature range of RT-750 .

Response: In the revised manuscript, we have added the TG curve of graphene from room temperature to 800 ℃ in the Figure 6. From the figure, we can see that the graphene is very stable, which changes little with the increase of temperature. Whereas, the TG curve of BP/G composite material has a significantly weight loss after 400 ℃, this is because that the phosphorus is volatilized under a high temperature and the residue is carbon. At the temperature of 800 ℃, the BP/G composite material has a residue of 42%, which indicates that the graphene is approximately 42% in the BP/G composite material. The data of TG results (Figure 6) has already been prolonged to 800℃, and we can see that the trend of data becomes flat and near 42%, which shows little difference with the previous result. However, the phosphorene is very thin and easy to be oxidized in the process of TG test, which cannot present the typical property. Thus, we can refer to the literature [35, 37], which indicates that the phosphorus can be volatilized under a high temperature.

5. Page 6. The authors declared that the fractured surface of WPU is smoothbut I think this is not the case. Please revise the description.

Response: Thank you for your instructive suggestions. We have improved the description in the revised manuscript. The detailed description is “The images of pure WPU shows homogeneous and without any additive particles”.

6. Page 7. The authors contributed little red spots in the pure WPU and G/WPU to the test error of the test or phosphorus element existed around the environment. But I think that phosphorus mapping of a region in G/WPU is obvious compared to pure WPU. How do you explain the results.

Response: We have added the corresponding EDS images in the Figure 8, we can see that there are little red spots in the pure WPU and G/WPU, whereas, the corresponding EDS results indicate that there is no phosphorus actually. Through communication with the professional analysis tests persons, the reason is given as the little phosphorus element existed around the environment or the interference signal from the peak near the position of phosphorus. In addition, the determination data is lower than the detectable limit of the EDS.

7. Page 9. The char residues for G/WPU is 0? Graphene has high char residues at high temperature. At least graphene is left after combustion.

Response: Thanks for your careful inspection, we did make a mistake and present a wrong data due to our negligence. The char residue for G/WPU is 4.0%. We have revised the data in the table 2.

8. The paper is hard to read due to the poor language, including plenty of grammar errors. Please revise the language thorough the text.

Response: Based on your kind suggestion, we have thoroughly revised the grammatical issues in our manuscript and inspected the format errors in detail, which will make our description more precise.

Round  2

Reviewer 2 Report

The authors have addressed all the issues the reviewers raised. I think the manuscript might be accepted in current form.